# Household and area determinants of emergency department attendance and hospitalisation in people with multimorbidity: a systematic review

Clare MacRae [1], Harry William Fisken,[2] Edward Lawrence,[3] Thomas Connor,[2] Jamie Pearce,[4] Alan Marshall,[5] Andrew Lawson,[6] Chris Dibben,[7] Stewart W Mercer [1], Bruce Guthrie [1]

For numbered affiliations see end of article.

**Correspondence to**
Dr Clare MacRae;
clare.macrae@ed.ac.uk

## ABSTRACT

**Objectives** Multimorbidity is one of the greatest challenges facing healthcare internationally. Emergency department (ED) attendance and hospitalisation rates are higher in people with multimorbidity, but most research focuses on associations with individual characteristics, ignoring household or area mediators of service use.

**Design** Systematic review reported using the synthesis without meta-analysis framework.

**Data sources** Twelve electronic databases (1 January 2000–21 September 2021): MEDLINE/OVID, Embase, Global Health, PsycINFO, ASSIA, CAB Abstracts, Science Citation Index Expanded/ISI Web of Science, Scopus, Cumulative Index to Nursing and Allied Health Literature, Sociological Abstracts, the Cochrane Library, and OpenGrey.

**Eligibility criteria** Adults aged ≥16 years, with multimorbidity. Exposure(s) were household and/or area determinants of health. Outcomes were ED attendance and/or hospitalisation. The literature search was limited to publications in English.

**Data extraction and synthesis** Independent double screening of titles and abstracts to select relevant full-text studies. Methodological quality was assessed using an adaptation of the Newcastle-Ottawa Quality Assessment Scale tool. Given high study heterogeneity, narrative synthesis was performed.

**Results** After deduplication, 10 721 titles and abstracts were screened, and 142 full-text articles were reviewed, of which 10 were eligible for inclusion. In people with multimorbidity, household food insecurity was associated with hospitalisation (OR 1.58 (95% CI 1.06 to 2.36) in concordant multimorbidity). People with multimorbidity living in the most versus least deprived areas attended ED more frequently (8.9% (95% CI 8.6 to 9.1) in most versus 6.3% (95% CI 6.1 to 6.6) in least), had higher rates of hospitalisation (26% in most versus 22% in least), and higher probability of hospitalisation (6.4% (95% CI 5.8 to 7.2) in most versus 4.2% (95% CI 3.8 to 4.7) in least). There was non-conclusive evidence that household income is associated with ED attendance and hospitalisation. No statistically significant relationships were found between marital status, living with others with multimorbidity, or rurality with ED attendance or hospitalisation.

## STRENGTHS AND LIMITATIONS OF THIS STUDY

⇒ Comprehensive literature search of 12 electronic databases to examine associations between household or area context and hospital attendance, specifically in people with multimorbidity
⇒ Reporting as per Preferred Reporting Items for Systematic Reviews and Meta-Analysis and Synthesis Without Meta-analysis guidelines
⇒ Limited evidence available in current literature with a small number of studies meeting eligibility criteria
⇒ Heterogeneity in multimorbidity definition and measurement, study exposures, outcomes, and analysis methods, precluded meta-analysis

**Conclusions** There is some evidence that household and area contexts mediate associations of multimorbidity with ED attendance and hospitalisation, but firm conclusions are constrained by the small number of studies published and study design heterogeneity. Further research is required on large population samples using robust analytical methods.

**PROSPERO registration number** CRD42021283515.

## INTRODUCTION

Multimorbidity is usually defined as the presence of two or more long-term physical or mental health conditions and is one of the greatest challenges facing healthcare internationally. It is becoming more common globally because of population ageing,[1 2] improved survival after acute illnesses such as myocardial infarction,[3] and changing patterns of mental health in younger people.[4] There are large independent associations between multimorbidity and adverse outcomes including frailty,[5] reduced functional health status,[6] and hospital attendance.[7] People with multimorbidity are at nearly three times the risk of hospitalisation compared with those without.[7]

Research usually compares different groups of people with multimorbidity, for example, defined by their individual characteristics, such as age, sex, and lifestyle factors.[8] [9] However, for this important patient group, social and environmental contexts are rarely examined in depth. People live in a social context, where attitudes and practices can be influenced by those around them; for example, households have been described as 'a living and developing unit of interdependent members, sharing common internal and external conditions and interactions'.[10] Research examining general adult populations (not stratified by multimorbidity) has found that people who live in a household with fewer financial or food resources,[11] [12] and single person households,[13] are at higher risk of emergency department (ED) attendance and hospitalisation. Likewise, the areas in which people live can influence health, and people living in urban areas and in areas with low socioeconomic position (SEP), with associated additional health burdens, are similarly at risk.[14] Households and areas are therefore an important unit of measurement when considering patterns of disease and healthcare use.[15] To date, there is a paucity of synthesis of studies examining these data in people with multimorbidity. Additionally, consideration of factors such as geographical proximity to services, known to be associated with likelihood of ED attendance,[16] and consideration of variation in how services themselves operate, including supply-side factors such as relative accessibility of primary care versus ED care, is important.

Understanding whether and how context influences outcomes is needed to ensure that the call by the Academy of Medical Sciences for research into 'how to organise healthcare systems to better manage patients with multimorbidity' is based on appropriate understanding of both the individual and their social and environmental context.[2] The aim of this review is to examine, in people with multimorbidity, associations of household and area contextual exposures with ED attendance and hospitalisation.

## METHODS
Method development and reporting of findings of the systematic review were based on the Preferred Reporting Items for Systematic Review and Meta-Analysis Protocols 2020 checklist.[17] The protocol was registered with PROSPERO CRD42021283515.

### Eligibility criteria and inclusion
Eligible studies included populations of adults aged 16 years and over, from the general population, who had been assessed for, and had, the presence of multimorbidity or multiple long-term conditions (LTCs), defined according to the Academy of Medical Sciences core definition of multimorbidity as 'two or more chronic conditions' (table 1).[2] Study exposure(s) examined were one or more household and/or area determinant of health, including built environment and socioeconomic determinants of

health (table 1).[18] Study outcomes were ED attendance and/or planned or unplanned hospitalisation.

### Search strategy
Twelve electronic databases (MEDLINE/OVID, Embase, Global Health, PsycINFO, ASSIA, CAB Abstracts, Science Citation Index Expanded/ISI Web of Science, Scopus, Cumulative Index to Nursing and Allied Health Literature, Sociological Abstracts, the Cochrane Library, and OpenGrey) and reference lists were searched for full-text articles meeting eligibility criteria, published between 1 January 2000 and 21 September 2021, which was the date the final searches took place. Searches were limited to articles written in English. An empirical approach to deriving the search terms was followed using a test set of three critical papers, one examining the household and area determinants of multimorbidity,[19] a second examining the definition and operationalisation of the term multimorbidity,[20] and a third meeting the study inclusion and exclusion criteria.[21] Key search terms, derived through reference to medical subject heading terms in test papers, iterated within MedLine to maximise sensitivity and specificity for relevant articles, were divided into three sets pertaining to multimorbidity, contextual exposures, and healthcare use (online supplemental table S1).

Following removal of duplicate records, independent double screening for relevance was performed for all titles and abstracts, using Covidence software,[22] and relevant full-text studies were selected (figure 1). Conflicts were resolved through discussion between screeners (CM, HWF, EL, and TC) at each stage of screening and eligibility of all included studies was verified by a fifth author (BG).

### Quality assessment
The Newcastle-Ottawa Quality Assessment Scale tool[23] was adapted (online supplemental box S1), and study quality screening for each paper was performed independently by two of the researchers (CM plus one of HWF, EL, and TC), with consensus achieved through discussion.

### Data synthesis
CM performed data extraction using a custom spreadsheet to record study design, location, population (age, definition of multimorbidity), exposures (household and/or area), and outcome measures (ED attendance and/or hospitalisation). If data were missing or unclear, we contacted corresponding authors for clarification who provided additional data and information regarding study methodology and analyses.

Study methodologies, exposures, outcomes, and effect measures were diverse and heterogeneous, and narrative synthesis was therefore performed according to Synthesis Without Meta-analysis[24] and PRISMA guidelines.[17] Studies were grouped according to exposure examined to ensure meaningful presentation of reviewed evidence.[25] Presentation of results using a standardised metric was not possible given the heterogeneity of outcome measures.

**Table 1** Study inclusion and exclusion criteria

| | Inclusion | Exclusion |
|---|---|---|
| Population | Adult participants from the general population residing in the community, aged 16 years and older and assessed for the presence of multiple LTCs (multimorbidity) | Participants initially selected based on the presence of index diseases, including any study examining comorbidity<br><br>Studies exclusively examining children aged 15 years and younger<br>Participants within hospital settings, or examination of readmission where the denominator is previously hospitalised participants |
| Exposure | ≥ 1 household- or area-level socioeconomic determinant of health (SDoH) in alignment with the WHO Commission on SDoH (CSDH) Framework[18]<br><br>Household: one residential unit, characteristics can include reference to the built environment or the occupants living within<br><br>Area: geographical area within which a person lives including all area sizes larger than a household unit | Individual SDoH only (e.g., ethnicity)<br>Study exposure(s) are direct 'causes' of ill-health, such as health behaviours (e.g., smoking) |
| Comparator | Study reports comparator group(s) for SDoH exposures (e.g., prevalence of hospitalisation admission in lowest versus highest household income) | Study does not report a comparator group for SDoH exposure(s) |
| Outcome | Prevalence or incidence studies examining emergency department use and hospitalisation (defined as a planned or unplanned overnight admission) | Studies not examining emergency department use or hospitalisation |
| Study design | Peer-reviewed studies of quantitative research designs (cross-sectional and longitudinal) | Systematic reviews, meta-analyses, clinical trials, and qualitative research |

LTCs, long-term conditions.

Due to the small number of included studies, all study results were reported with quantitative figures given for statistically significant findings. Narrative comparison of heterogeneity between studies examining each exposure was performed by reviewing results within each group, comparing methodologies, exposures, and outcomes.

### Patient and public involvement

Patients or the public were not involved in the design, conduct, reporting, or dissemination plans of this research.

### RESULTS

The systematic search identified 16853 articles. After removal of 6132 duplicates, 10721 titles and abstracts were screened, of which 10579 were excluded at the first screening stage. Full-text screening of 142 articles identified 10 studies meeting eligibility.

### Study characteristics

Three studies were conducted in Asia,[26–28] three in Europe,[21 29 30] two in North America,[31 32] one in Africa,[33]

and one used multiple sites[34] (table 2). All studies used cross-sectional data and of these, two used data derived from existing cohort studies.[31 32] Study sample size ranged from 1670 to 5316830 (median 27689 participants, interquartile range (IQR) 20689–216633). Seven studies reporting regression model outcomes,[21 26–28 31 33] used smaller populations (1670–162464 participants, median 24642, IQR 15387–66474), than the two studies reporting rates of hospital attendance (2262698–5316830, median 3789764, IQR 3026231–4553297).[29 32] Each included study examined a specific age group, which were ≥15 years old (one study),[28] 15–65 years old (one study),[26] ≥18 years old (two studies),[27 32] ≥20 years old (two studies),[30 34] ≥40 years old (one study),[33] ≥50 years old (one study),[21] and ≥65 years old (two studies).[29 31]

Multimorbidity was defined as the presence of two or more LTCs by all studies, with the number of LTCs included in multimorbidity measures ranging from 8 to 52 (median 26.5, IQR 13–41.5). Nine studies included both physical and mental LTCs,[21 27–35] and one study only included physical LTCs.[26] Various approaches were taken to defining the list of LTCs included, ranging from no

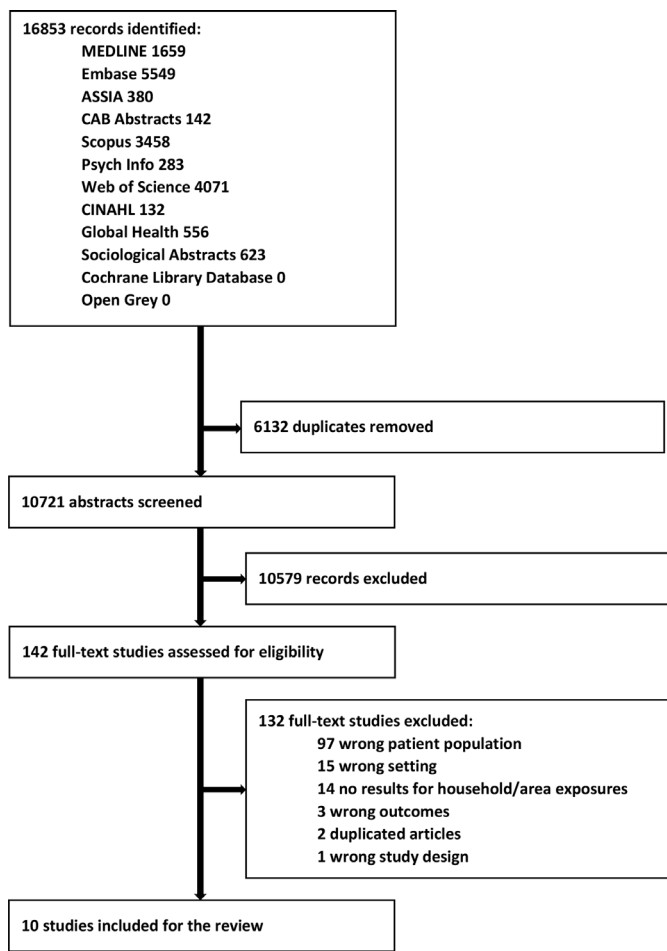

16853 records identified:
  MEDLINE 1659
  Embase 5549
  ASSIA 380
  CAB Abstracts 142
  Scopus 3458
  Psych Info 283
  Web of Science 4071
  CINAHL 132
  Global Health 556
  Sociological Abstracts 623
  Cochrane Library Database 0
  Open Grey 0

6132 duplicates removed

10721 abstracts screened

10579 records excluded

142 full-text studies assessed for eligibility

132 full-text studies excluded:
  97 wrong patient population
  15 wrong setting
  14 no results for household/area exposures
  3 wrong outcomes
  2 duplicated articles
  1 wrong study design

10 studies included for the review

**Figure 1** Study selection, PRISMA diagram. PRISMA, preferred reporting items for systematic reviews and meta-analyses.

included rationale by Lu et al[26] and Stafford et al,[21] adaptation of previous studies examining multimorbidity by Oureta et al,[29] to including LTCs included in the Charlson and Elixhauser indexes by Mbuya-Bienge et al.[32] Tomita et al[33] examined both concordant and discordant multimorbidity, fitting separate logistic regression models for each. In this study, discordant multimorbidity was defined as the occurrence of LTCs in more than one domain in mental health, non-communicable, and communicable disease.

The 10 included studies examined six unique exposures, where an exposure was defined as any household or area characteristic associated with ED attendance or hospitalisation. Seven studies examined household,[21 26–28 31 33 34] and four studies examined area exposures.[27 29 30 32] One study, by Pati et al,[27] examined both household and area exposures.

Household exposures examined were household income (three studies),[28 31 34] multimorbidity status of other household members (one study),[21] household food insecurity (one study),[33] and marital status (four studies).[26–28 33] Data on household income were derived through household surveys in all included studies.[28 31 34] Household food insecurity was defined

using a cross-cultural measure, where survey participants were asked how often there was no food in their household and how often household members went to sleep hungry or did not eat for a day.[33] Multimorbidity status of other household members was determined through linkage of local government and health provider administrative data with the unique property reference number,[21] a unique identifier for every addressable location in the UK.[36] Marital status was the most examined household exposure, but studies varied in how they examined this. Pati et al[27] classified participants as currently married and currently not married, and Lu et al[26] defined groups as married and single/divorced. Tomita et al[33] and Chung et al[28] divided participants into three groups comparing those who are currently married with never married or separated/divorced/widowed. Stafford et al[21] examined multimorbidity status of household the other household resident in two-person households, with no reference to relationship status.

Area exposures examined were area SEP (three studies)[29 30 32] and rurality (one study).[27] All studies examining area SEP used a similar approach to deriving their SEP measure, using quintiles of a score derived from census tract deprivation indexes, based on rates of unemployment, educational attainment, and type of employment. Rurality was examined as a dichotomous exposure of rural versus urban, derived from population size and density, and proportion of workers.

Reported outcomes were any ED attendance (two studies),[31 32] any hospitalisation (whether planned or unplanned, and some studies did not differentiate whether hospitalisations were planned or unplanned) (six studies),[26 28 29 31 33 34] any inpatient or outpatient hospital attendance (one study),[27] any ED attendance or unplanned hospitalisation (one study),[21] and any unplanned or emergency hospitalisation or unplanned or emergency potentially avoidable hospitalisation (one study).[30] No studies presented results stratified by the number of attendances in the study period.

Methodological quality of studies varied across studies and was rated high in three studies[30 31 33] and medium in six[21 26 27 29 32 34] (online supplemental table S2). Potential bias related to reporting of hospital attendance outcomes, where studies used either questionnaire-based self-reported outcomes[26 27 33 34] or electronic health records and health payment systems[21 29 31 32]; therefore, differences in accuracy and reporting could exist between these studies. Use of statistical reporting varied between studies, with six studies performing regression models with different model outcomes, including odds ratios (ORs),[21 26 31 33] relative risk (RRs),[28] incidence rate ratios (IRRs),[27] event probability,[34] as well as studies examining unadjusted rates of hospital attendance.[29 32]

## Household income

There was inconclusive evidence of a relationship between household income and hospital use. Chung et al,[28] a medium quality study from Hong Kong, examined

**Table 2** Study characteristics

| Author | Study purpose | Study design | Data source | Sample size | Setting and date | Study population | Multimorbidity definition | Household exposure | Area exposure | Primary outcome measure(s)* |
|---|---|---|---|---|---|---|---|---|---|---|
| Chung et al (2016), Hong Kong[28] | Examine factors associated with hospitalisation | Cross-sectional | Hong Kong Government household survey | 25780 | General population, 2011 | ≥15 years | ≥ 2 LTCs out of 45 LTCs, mental and physical | Household income and marital status | NA | 12-month hospitalisation† |
| Fisher et al (2021), Canada[31] | Examine sociodemographic and health factors associated with acute care | Cross-sectional | Canadian Community Health Survey; administrative health data | 28361 | General population, pooled from multiple cycles 2005–2012 | 65–85 years | ≥ 2 LTCs out of 12 LTCs, physical and mental | Household income | NA | 12-month ED attendance and hospitalisation† |
| Lu et al (2017), China[26] | Examine inequalities limiting utilisation of health services | Cross-sectional | China Labour Force Dynamic Survey | 23505 | Adult population in employment, 2014 | 15–65 years | ≥ 2 LTCs out of 10 LTCs, physical only | Marital status | NA | 12-month hospitalisation† |
| Mbuya-Bienge et al (2021), Canada[32] | Examine SEP and frequency of use of healthcare | Cross-sectional | Quebec Integrated Chronic Disease System | 5316830 | General population, 2011–2012 | ≥18 years | ≥ 2 LTCs out of 31 LTCs, physical and mental | NA | Area SEP | 12-month ED attendance |
| Orueta et al (2013), Spain[29] | Examine costs of multimorbidity by SEP | Cross-sectional | Routine administrative health data | 2262698 | General population, 2011 | ≥65 years | ≥ 2 LTCs out of 52 LTCs, physical and mental | NA | Area SEP | 12-month hospitalisation† |
| Pati et al (2015), India[27] | Examine relationship between multimorbidity and healthcare utilisation | Cross-sectional | Multistage random sampling patient interviews | 1670 | General population, 2013–2014 | ≥18 years | ≥ 2 LTCs out of 22 LTCs, physical and mental | Marital status | Rurality | 12-month hospital inpatient or outpatient hospital use |
| Payne et al (2013), Scotland, China, Hong Kong[30] | Examine association between SEP and unplanned hospitalisation | Cross-sectional | Scottish Practice Team Information dataset | 180815 | General population, 2006 | ≥18 years | ≥ 2 LTCs out of 52 LTCs, physical and mental | NA | Area SEP | 12-month emergency or potentially preventable hospitalisation |
| Stafford et al (2020), England[21] | Examine association between co-resident multimorbidity status and hospitalisation | Cross-sectional | Barking and Dagenham (B&D) health data, Clinical Practice Research Datalink (CPRD) | 9222 (B&D) and 10528 (CPRD) | General population, 2016–2018 | ≥ 50 years living in two-person households | ≥ 2 LTCs out of 16 LTCs, physical and mental | Multimorbidity status of other household member | NA | Non-elective hospitalisation |
| Tomita et al (2021), Tanzania[33] | Examine association between SEP and hospitalisation | Cross-sectional | Dar es Salaam Health Demographic Surveillance System | 2299 | General population, 2017–2018 | ≥ 40 years | ≥ 2 LTCs out of 8 LTCs (concordant and discordant)‡, physical and mental | Marital status Household food insecurity | NA | 12-month hospitalisation† |
| Wang et al (2015), Scotland, China, Hong Kong[34] | Examine association between household income with hospitalisation | Cross-sectional | Scotland: Health Surv | 36921 | General population, 2008 | ≥ 40 years | ≥ 2 LTCs out of 31 LTCs, physical and menta | Household income | NA | 12-month hospitalisation† |
| | | | China:Routine administrative data | 162 464 | 2011 | | | | | |
| | | | Hong Kong:Household Surve | 29187 | 2011 | | | | | |

NA: not applicable

*All outcomes are dichotomous (attendance or no attendance) rather than counts.

†No differentiation planned/unplanned.

‡Concordant multimorbidity: one domain of mental health/non-communicable/communicable health. Discordant multimorbidity: ≥domains.

ED, emergency department; LTCs, long-term conditions; SEP, socioeconomic position.

the relative risk (RR) of hospitalisation for households with different levels of income. Analyses were compared with household income of <$10 000, and RRs were close to 1.00 for all income strata, with all 95% confidence intervals (95% CIs) crossing 1.00 apart from income of $10–19 000, RR 1.66 (95% CI 1.002 to 1.356). Fisher et al,[31] a Canadian study rated as high quality, examined associations between household income and service use in people with multimorbidity, with separate regression models fitted to subgroups defined by age-group (65–74 and 75–84 years), sex, multimorbidity (categorised as 2–3 and ≥4 LTCs), and outcomes (ED attendance and hospitalisation) (table 3). In people with multimorbidity, ORs for ED attendance in people with middle (Canadian $30-$79 900) and lower (<$30 000) household income, versus high (>$80 000), were almost all >1.00 (15 out of 16). However, many (12 out of 16) had wide 95% CIs crossing 1.00, which is likely to reflect the small numbers in each stratum. There was some evidence of a dose–response relationship (ORs for low versus high income were always larger than ORs for medium versus high income), and estimated ORs were all small to moderate in size (maximum likelihood in men aged 65–74 years with ≥4 LTCs was OR 2.74 (95% CI 1.12 to 6.66) for low versus high household income). In models with hospitalisation as outcome, the ORs for middle and lower household income, versus high, were all >1.00; however, only two (out of 16) ORs had 95% CIs not crossing 1.00, and a similar dose–response relationship between degree of income and hospitalisation was seen.

Wang et al,[34] a study rated as medium quality, found that in people with multimorbidity, lower household income was associated with a higher probability of hospitalisation in Scotland and public hospitalisation in Hong Kong. A dose–response relationship was seen where probability of admission rose as the number of LTCs increased, for example, probability for hospitalisation was higher in people with ≥4 LTCs (probability 30.7% (95% CI 30.3 to 31.7)) than people with two LTCs (probability 18.85% (95% CI 18.4 to 19.2)) in the lowest household income group in Scotland. However, a reversed relationship, lower household income associated with lower probability of hospitalisation, was seen in China and in private hospitals in Hong Kong. For example, people with lowest household income had lowest probability of hospitalisation (probability 24.7% (95% CI 23.4 to 26.1)) versus people with highest household income (probability 36.1% (95% CI 33.6 to 38.6)), in people with ≥4 LTCs with no medical insurance in China.

### Household coresident multimorbidity status
An English study by Stafford et al[21] examined the multimorbidity status of the other household resident in two-person households. In this study rated as medium quality, no significant difference between living with someone who had multimorbidity, versus someone who did not have multimorbidity, was found in association with ED attendance (OR 1.08 (95% CI 0.95 to 1.23)).

### Household hunger
A statistically significant relationship between severe household hunger and hospitalisation was found in a Tanzanian study by Tomita et al,[33] rated as high quality. This relationship was found in both concordant (severe versus none-to-moderate household hunger OR 1.58 (95% CI 1.06 to 2.36)) and discordant (severe versus none-to-moderate household hunger OR 1.54 (95% CI 1.04 to 2.28)) multimorbidity.

### Marital status
No significant association between marital status and any outcome was found in the four studies examining this. Chung et al[28] examined RR of hospitalisation depending on marital status, and there were no statistically significant results with all RR being close to 1.00 with wide 95% CIs crossing 1.00. Lu et al,[26] a Chinese study rated as medium quality, found a small but not statistically significant relationship between being married and hospitalisation (married versus unmarried OR 1.04 (95% CI 0.54 to 2.02)). Pati et al,[27] a study from India rated as medium quality, found a larger but not statistically significant relationship in the opposite direction between marital status and any inpatient or outpatient hospital attendance (unmarried versus married: IRR 1.17 (95% CI 0.85 to 1.61)]). Similarly, Tomita et al[33] found a small but not statistically significant association between not being married and hospitalisation in people with both concordant multimorbidity (never married versus married OR 1.43 (95% CI 0.62 to 3.28); divorced/separated versus married OR 1.24 (95% CI 0.86 to 1.7)) and in people with discordant multimorbidity (never married versus married OR 1.44 (95% CI 0.63 to 3.28); divorced/separated versus married OR 1.27 (95% CI 0.83 to 1.85)) in Tanzania.

### Socioeconomic position
More people living in the most versus least deprived quintile of areas attended ED in a medium quality Canadian study by Mbuya-Bienge et al[32] (table 4). Results were stratified by number of LTCs and in people with two LTCs, 8.9% (95% CI 8.6 to 9.1%) living in most deprived areas attended ED versus 6.3% (95% CI 6.1 to 6.6%) in the least deprived areas; for people with three LTCs 12.0% (95% CI 11.6 to 12.5%) versus 9.5% (95% CI 9.1 to 10.0%); and for people with more than four LTCs 18.3% (95% CI 17.9 to 18.8%) versus 16.8% (95% CI 16.3 to 17.4%). Likewise, more people living in the most versus least deprived area quintiles were hospitalised in the Spanish study by Orueta et al,[29] a study rated as medium quality. This study also stratified the number of LTCs and found that in people with four to six LTCs, 26% living in most deprived areas were hospitalised versus 22% of people in the least deprived areas; for people with seven to nine LTCs, 48% versus 44% (no 95% CIs or cohort size from which to derive these were provided). However, in people with ≥10 LTCs, rates of hospitalisation were lower in people living in the most, 68%, versus least, 70%,

**Table 3** Household exposure study results

| Study exposure | Study | Outcome | Methodological quality rating | Covariates | Result metric | Key results | Association at 95% for ORs, RRs, and IRRs* |
|---|---|---|---|---|---|---|---|
| Household income | Fisher et al Canada[31] | ED attendance and any hospital use | High | Stratified by age and number of LTCs | ORs and 95% CIs | Highly stratified sample with small group sizes 6 of 32 models showed statistically significant relationship between increased likelihood of ED attendance/any hospital use with lower household income | ✓/– |
| | Chung et al China[28] | Hospitalisation | Medium | No details provided | RR and 95% CIs | Highest RR of hospitalisation in households with income ≥$50 000 (RR 1.193 (0.916–1.553)) versus income <$10 000 (reference RR 1.000) | – |
| | Wang et al Scotland, Hong Kong, China[34] | Hospitalisation | Medium | Stratified by number of LTC and sex | Predicted probability (P) and 95% CIs | Scotland: higher probability of hospitalisation in lowest (18.7% (18.3 to 19.1)) versus highest (11.1% (10.7 to 11.4)) household income. China (no medical insurance): lower probability of hospital admission in lowest (8.2% (7.9 to 8.5)) versus highest (13.1% (12.2 to 14.0)) household income | NA |
| Coresident multimorbidity | Stafford et al England[21] | ED attendance | Medium | Age, sex, SEP | ORs and 95%CIs | No difference in ED attendance between people with household coresident with versus without multimorbidity (OR 1.08 (0.95 to 1.23) | – |
| Household food insecurity | Tomita et al Tanzania[33] | Hospitalisation | High | Age, sex, marital status, education | ORs and 95% CIs | Increased likelihood of hospitalisation in severe versus little-to-no household food insecurity (OR 1.58 (1.06 to 2.36)) | ✓ |
| Household marital status | Tomita et al Tanzania[33] | Hospitalisation | High | Age, sex, education | ORs and 95% CIs | No difference in hospitalisation between currently married versus never married (OR 1.43 (0.62 to 3.28)) or separated/divorced (OR 1.24 (0.86 to 1.78)) | – |
| | Chung et al China[28] | Hospitalisation | Medium | No details provided | RR and 95% CIs | No difference in RR of hospitalisation in widowed (RR 1.058 (0.791 to 1.416) or in married (RR 0.917 (0.706 to 1.191)) versus single (reference RR 1.000) | – |
| | Lu et al China[26] | Hospitalisation | Medium | No details provided | ORs and 95% CIs | No difference in hospitalisation between married or single/divorced (OR 1.04 (0.54 to 2.02)) | – |
| | Pati et al India[27] | Inpatient and outpatient hospital use | Medium | Age, sex, ethnicity, education, SEP, rurality | IRRs and 95% CIs | No difference in inpatient or outpatient hospital use between currently married versus currently not married (IRR 1.17 (0.85 to 1.61)) | – |

*'NA' study outcomes where no ORs, RRs, or IRRs were reported, '✓' studies where an association at 95% significance was found, '–' studies where no association at 95% significance was found, '✓/–' studies where results were mixed.
ED, emergency department; IRRs, incidence rate ratios; LTCs, long-term conditions; SEP, socioeconomic position.

## Table 4 Area exposure study results

| Study exposure | Study | Outcome | Methodological quality rating | Covariates | Result metric | Key results | Association at 95% for IRR* |
|---|---|---|---|---|---|---|---|
| Area SEP | Payne et al Scotland, Hong Kong, China[30] | Hospitalisation | High | Stratified by number of LTC and sex | Predicted probability and 95% CIs | Higher probability of unplanned hospitalisation in men with physical only multimorbidity in most (6.4% (5.8% to 7.2%)) versus least (4.2% (3.8% to 4.7%)) deprived areas Higher probability of unplanned hospitalisation in men with physical and mental multimorbidity in most (12.1% (11.0% to 13.6%)) versus least (8.1% (7.3% to 9.2%)) deprived areas | NA |
| | Mbuya-Bienge et al Canada[32] | ED attendance | Medium | Non-adjusted prevalence rates | Prevalence rate % and 95% CIs | Lower rates of ED use in most (8.9% 8.6% to 9.1%)) versus least (6.3% (6.1% and 6.6%)) deprived areas | NA |
| | Orueta et al Spain[29] | Hospitalisation | Medium | Non-adjusted prevalence rates | Prevalence rate % (no 95% CIs) | Lower rates of hospitalisation in most (26%) versus least (22%) deprived area quintile | NA |
| Rurality | Pati et al India[27] | Inpatient and outpatient hospital use | Medium | Age, sex, ethnicity, education, SEP and rurality | IRR and 95% CIs | No difference in hospitalisation between urban and rural areas (IRR 1.09 (0.64 to 1.88)) | – |

*'NA' study outcomes were not ORs or IRRs, '✓' studies where an association at 95% significance was found, '–' studies where no association at 95% significance was found, '✓/–' studies where results were mixed.
ED, emergency department; IRRs, incidence rate ratios; LTCs, long-term conditions; SEP, socioeconomic position.

deprived areas. Neither study examining the effect of SEP performed regression analyses or stratified by age.

Payne et al,[30] a Scottish study rated as high quality, examined the association between SEP and hospitalisation through regression analyses, stratified by sex, number of LTCs, and presence of physical only or physical and mental LTCs. Living in the most versus least deprived quintile areas was associated with a higher probability of hospitalisation for all groups examined (e.g., men with physical only multimorbidity had a higher probability of unplanned hospitalisation in most (6.4% (95% CI 5.8% to 7.2%)) versus least (4.2% (95% CI 3.8% to 4.7%)) deprived areas. These effects were more pronounced for people with more versus less LTCs, with a dose response seen between number of LTCs and increase in probability of hospitalisation, and for people with physical and mental multimorbidity versus physical only multimorbidity.

### Area rurality

Pati et al[27] examined the relationship between urban versus rural residence and any hospital attendance in India and found a relationship that did not reach statistical significance (urban versus rural residence IRR 1.09 (95% CI 0.64 to 1.88)).

## DISCUSSION
### Principal findings

This systematic review and narrative synthesis describe existing evidence of associations between household and area exposures with ED attendance and hospitalisation outcomes in people with multimorbidity. In people with multimorbidity, household food insecurity was associated with hospitalisation, and rates of ED attendance and hospitalisation, and probability of hospitalisation, were higher in people living in the most versus least deprived areas. There is non-conclusive evidence that in people with multimorbidity household income was associated with ED attendance or hospitalisation, with differing relationships seen depending on study location and how healthcare is organised and paid for in different settings. No statistically significant relationships were found in people with multimorbidity between ED attendance or hospitalisation with marital status, living with others with multimorbidity, or living in urban versus rural areas.

### Strengths and weaknesses

Strengths of the study include comprehensive and systematic searching of many electronic databases, with an iteratively developed set of search terms, extraction of data by two researchers and assessment of study quality with adaptation of a standard method to meet the needs of the review and analysis using a standardised approach to narrative reporting. There are several limitations of our study. There are few included studies because analyses of household and area variables associated with ED attendance or hospitalisation have not been commonly done in people with multimorbidity, and there is considerable

heterogeneity in the definition and measurement of multimorbidity, exposures, outcomes, and analysis methods, precluding meta-analysis. Studies were from high-, low-, and middle-income countries, where contextual factors may hold different meanings, making direct comparisons across studies difficult. Finally, although multimorbidity was defined as the presence of two or more LTCs in all studies, there was a diverse range of LTCs included within multimorbidity measures, which will affect prevalence rates and therefore reliability of comparison between studies.[20]

### Comparison with existing literature

Household income was not conclusively associated with ED attendance or hospitalisation in people with multimorbidity; however, small sample sizes resulting from stratified analyses may have contributed to some estimates not reaching statistical significance in the Canadian study examining this.[31] Different directions of association were found between household income and hospitalisation in different countries and healthcare systems by Wang et al.[34] Lower household income was associated with higher likelihood of hospitalisation in areas with universal healthcare provision (free at the point of care in Scotland and heavily subsidised in Hong Kong public hospitals). These results reflect findings from a study of older people (not stratified by multimorbidity) in Stockholm, where healthcare is free at the point of delivery, that found women living in the lowest versus highest income households had an increased probability (OR 2.91 (95% CI 2.52 to 3.36)) of ED attendance.[37] However, lower household income was associated with lower probability of hospitalisation in areas where healthcare is largely or solely privately funded (mainland China and the private hospitals in Hong Kong), demonstrating that lack of access to equitable care can further exacerbate health inequalities by reducing access to those with lowest household income.[34]

Household hunger was associated with hospitalisation in both concordant and discordant multimorbidity in Tanzania,[33] which is similar to a study from the USA where household hunger was associated with increased hospitalisations in a general adult population (OR 1.36 (95% CI 1.22 to 1.52)).[38] The severity of household hunger is likely to vary widely depending on study setting, making these findings context specific. Marital status in this study was assumed to be a household-level variable, but it can also be considered a characteristic of the individual. It is commonly examined as a proxy for not living alone and having increased household social support in studies using routine data,[12 39] and therefore in this study was assumed to be a household-level variable. However, actual living arrangements (including cohabitation as an unmarried couple) are more closely associated with health outcomes but are less examined due to limitations in data availability.[39] We expected to see a protective effect between being married resulting in reduced ED attendance and hospitalisation, but the included studies found no statistically significant associations in either direction. The three studies examining area SEP reported increased ED attendance and hospitalisation in people with multimorbidity living in the most deprived fifth of areas, consistent with existing literature concerning general adult populations where people living in areas of lower area SEP are higher users of healthcare.[8 9 40]

One study in our review examined the association between living in urban versus rural areas and any hospital attendance in people with multimorbidity, finding a non-statistically significant OR >1.00 for people living in urban areas. A study examining a population of older adults in China,[41] not stratified by multimorbidity status, found that living in urban areas was significantly associated with increased likelihood of inpatient care.

Some of the variation in results across studies may be explained by the heterogeneity in the definition of multimorbidity. Future studies should standardise multimorbidity definitions (e.g., the presence of two or more LTCs from a standardised list of LTCs) to align with other studies and improve comparability.[20]

### Implications

Comparative research that further explores variation in associations in different healthcare systems would be valuable, testing models of exposure to outcome associations. Such studies are needed to understand the mechanisms by which observed associations happen, examining whether household and area exposures interact with each other and with multimorbidity, considering whether these factors mediate, moderate, or have a multiplicative effect. No included articles examined supply factors associated with hospital use, for example, area wide demand issues such as distance to services, or supply issues such as access to primary care services. These factors are likely to influence hospital use rates in people with multimorbidity and require further study. Use of longitudinal study designs could be used to evaluate the evolution of increased risk or protection associated with the context in which an individual lives. Only two area-level exposures have been examined, despite more extensive literature examining the place-based determinants of multimorbidity,[19] emphasising the need to examine associations between outcomes in people with multimorbidity and a broader range of area-level factors, including green spaces, social cohesion, and provision of services.

### Conclusions

In conclusion, although available evidence in the study area is limited, there is some evidence that household and area exposures are associated with increased risk of ED attendance and hospitalisation in people with multimorbidity. Since interventions to reduce hospital attendance may be more effective if they also account for the context in which people live, there is a need for further research to examine the contribution of a wider range of contextual exposures to hospital attendance in people with multimorbidity.

**Author affiliations**
¹Usher Institute of Population Health Sciences and Informatics, The University of Edinburgh, Edinburgh, UK
²The University of Edinburgh Edinburgh Medical School, Edinburgh, UK
³Ninewells Hospital and Medical School, Dundee, UK
⁴Institute of Geography, University of Edinburgh Institute of Geography, Edinburgh, UK
⁵Department of Social Policy, The University of Edinburgh Social Policy, Edinburgh, UK
⁶Department of Public Health Sciences, Medical University of South Carolina, Charleston, South Carolina, USA
⁷Institute of Geography, University of Edinburgh, Edinburgh, UK

**Contributors** CM and BG conceived the study and contributed to the design of the project. CM created the search strategy, and BG, SWM, JP, AL, and CD contributed to subsequent iterations. CM, HWF, EL, and TC conducted the search. CM and BG extracted and analysed the data. CM drafted the paper, and BG, SWM, JP, AM, AL, and CD critically commented on the manuscript. All authors reviewed drafts and approved the final version. CM is the study guarantor and accepts full responsibility for the work and/or the conduct of the study, had access to the data, and controlled the decision to publish

**Funding** CM received funding from the National Health Service (NHS) Education for Scotland Academic Fellowship and Medical Research Council MR/W000253/1 to support this work.

**Competing interests** None declared.

**Patient and public involvement** Patients and/or the public were not involved in the design, or conduct, or reporting, or dissemination plans of this research.

**Patient consent for publication** Not applicable.

**Ethics approval** Not applicable.

**Provenance and peer review** Not commissioned; externally peer reviewed.

**Data availability statement** No data are available. No additional data are available. No data sharing agreement was obtained for this study because data analysed were obtained from previous published studies where data sharing agreements are in place.

**ORCID iDs**
Clare MacRae http://orcid.org/0000-0002-1007-683X
Stewart W Mercer http://orcid.org/0000-0002-1703-3664
Bruce Guthrie http://orcid.org/0000-0003-4191-4880

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
