## [Reviewer comments · BMJ Open]

ARTICLE DETAILS

TITLE (PROVISIONAL)	Household and area determinants of emergency department attendance and hospitalisation in people with multimorbidity: a systematic review
AUTHORS	MacRae, Clare; Fiskien, Harry; Lawrence, Edward; Connor, Thomas; Pearce, Jamie; Marshall, Alan; Lawson, Andrew; Dibben, Chris; Mercer, Stewart; Guthrie, Bruce

VERSION 1 – REVIEW

REVIEWER	Jessica Sheringham University College London
REVIEW RETURNED	27-Apr-2022

GENERAL COMMENTS	Thank you for inviting me to review this paper. I have reviewed it in collaboration with Dr Elizabeth Ingram, UCL and the comments represent both our views on the paper. We commend the authors on a well written review in general and consider it addresses the important topic of how household and area determinants are associated with emergency department attendance and hospitalisation in people with multimorbidity. We had a number of comments that should be addressed before publication: Major 1. There is little discussion (either in the introduction or discussion) around how demand and supply-side factors influence health and care use. For example, an individual may choose or not choose to use hospital services based on demand-side determinants like distance between their residence and a hospital, transport links, opening times of the service, and the decision of primary care clinicians or paramedics. However, decisions about emergency department use can also be affected by supply-wide issues, e.g. lack of primary care access or in contrast some ED attendance/emergency hospitalisation can result from a referral to from primary care rather than being patient-initiated. A more indepth consideration of this could have implications for the paper in several ways, for example: - In the methods, data extraction could also include the covariates that study authors included in their adjusted analysis to understand whether any of these factors could explain the findings- In the discussion it may enable more informative comment around why the authors think the results reported may have been found (given covariates adjusted for). At present, the discussion of this paper in points feels quite repetitive of the results.
---

	2. Clearer definition of the outcome: The paper's hospitalisation outcome as we understand it is very broad. Does it include planned and unplanned hospitalisations, day visits and overnight admissions? Some clarity around the definition of the outcome of interest and breaking down as it has been used in the papers at least into elective and non-elective could have implications for the findings. 3. Results: we are not statisticians, but we would have interpreted the results of the Fisher paper differently. The results presented for Fisher et al.'s study are based on models that include interaction terms and, as such, we believe the more suitable analyses to present are found in Table 3 of the paper (which show there is a statistically significant relationship between household income and hospitalisation rates). In addition, some clarification as to why Fisher et al.'s results for two determinants - living arrangement and rurality - aren't included in this review would be helpful. 4. Paper selection: could you clarify why two further papers do not meet the inclusion criteria as we feel these papers would be relevant for inclusion given this review's inclusion criteria:  - Chung et al., (2016) The association between types of regular primary care and hospitalization among people with and without multimorbidity: A household survey on 25,780 Chinese - Sum et al., (2020) Implications of multimorbidity on healthcare utilisation and work productivity by socioeconomic groups: Cross-sectional analyses of Australia and Japan 5. Clarity of findings tables. We both struggled to understand what the main findings of the papers were in the findings tables. Some suggestions are given:  - Reporting of the covariates adjusted for in each study, as these will likely influence study results. - The results might also be more digestible if the authors included subheadings (for example study characteristics, key results for household income studies, then area deprivation studies, then rurality and so on) - Inclusion in Table 3 of a pictorial measure so the reader can easily see the results of each study at a quick glance, rather than having to read sentences - Inclusion of quality ratings in the results table so reader can understand results alongside quality assessments. Minor comments/questions that might require some clarification or revisions:  1. More information would be useful on the search strategy: On what dates were the database searches conducted and what did the authors do about non-English papers? 2. Sometimes useful to clarify why a variable is a household level exposure: marital status for example, could be considered an individual level exposure, but possibly it's used as a proxy for living with a partner? 3. It would be useful to draw out the strengths and limitations of this review rather than the literature included in the review e.g., the fact there's little literature and multimorbidity is differentially defined isn't a limitation of the review itself but of the body of literature. There are a few additional strengths worth noting (e.g., considerable use of second screening, multiple people extraction data and assessments
--	--

	of quality using a tailored version of the NOS). There are also a few additional limitations worth noting (no meta-analysis, mix of high income and low-middle income countries in which contextual factors may hold different meanings across contexts and exclusion of non-English papers)
REVIEWER	Alexandru Dregan King's College London, UK, Primary Care and Public Health Sciences
REVIEW RETURNED	11-Jun-2022
GENERAL COMMENTS	The authors have done an excellent work at summarising current gaps in the evidence between contextual determinants of hospitalisation with multimorbidity. Though the review does not include a meta-analysis (justified accordingly by the authors), it provides valuable information on a possible role (albeit unclear at the moment) of area-level circumstances on the growing burden of multimorbidity on the health care system. The study is methodologically appropriate and acknowledges the complexities of drawing overall conclusions from studies employing diverse measures of multimorbidity and hospitalisation. The manuscript is well-written and should be of interest to the journal audience. My only minor comment would be to include the study country in the tables (include study exposure as a row subheading?) - might be informative with regards to some of the inconsistent evidence for countries with different distribution of wealth/income inequalities (e.g., China vs India or Canada).

VERSION 1 – AUTHOR RESPONSE

Reviewer: 1

Ms. Jessica Sheringham, University College London

Comments to the Author:

Thank you for inviting me to review this paper. I have reviewed it in collaboration with Dr Elizabeth Ingram, UCL and the comments represent both our views on the paper.

We commend the authors on a well written review in general and consider it addresses the important topic of how household and area determinants are associated with emergency department attendance and hospitalisation in people with multimorbidity. We had a number of comments that should be addressed before publication:

Major

1. There is little discussion (either in the introduction or discussion) around how demand and supply-side factors influence health and care use. For example, an individual may choose or not choose to use hospital services based on demand-side determinants like distance between their residence and a hospital, transport links, opening times of the service, and the decision of primary care clinicians or paramedics. However, decisions about emergency department use can also be affected by supply-wide issues, e.g. lack of primary care access or in contrast some ED attendance/emergency hospitalisation can result from a referral to from primary care rather than being patient-initiated.

Response 4: Thank you for making this useful point. We have added additional text to reference these issues in the article introduction, page 5, lines 23-26:

“Additionally, factors such as geographical proximity to services are known to be associated with likelihood of ED attendance,¹⁵ and consideration of individual characteristics and variation in how services themselves operate, including supply-wide factors such as relative accessibility of primary care versus ED care are important.”

Also, in the article discussion Page 20, lines 18-21:

“No included articles examined supply factors associated with hospital use, for example area wide demand issues such as distance to services, or supply issues such as access to primary care services. These factors are likely to influence hospital use rates in people with multimorbidity and require further study.”

A more in-depth consideration of this could have implications for the paper in several ways, for example:

- In the methods, data extraction could also include the covariates that study authors included in their adjusted analysis to understand whether any of these factors could explain the findings
- In the discussion it may enable more informative comment around why the authors think the results reported may have been found (given covariates adjusted for). At present, the discussion of this paper in points feels quite repetitive of the results.

Response 5: We have added an additional “Covariates” column to Table 3 and Table 4 including this information. The majority of results had small effect sizes or were not statistically significant. Addition of covariates to these models may have had some effect on the results, however small study sizes are more likely to be responsible for the lack of statistical significance.

Please see page 19, lines 19-21, where we have added additional discussion to explain the findings:

“demonstrating that lack of access to equitable care can further exacerbate health inequalities by reducing access to those with lowest household income.³⁴”

2. Clearer definition of the outcome: The paper’s hospitalisation outcome as we understand it is very broad. Does it include planned and unplanned hospitalisations, day visits and overnight admissions? Some clarity around the definition of the outcome of interest and breaking down as it has been used in the papers at least into elective and non-elective could have implications for the findings.

Response 6: Please see Table 2, “Primary outcome(s) measures” for details of each study outcome(s). The outcome definition includes emergency department attendance and hospital

admission, and most included studies did not provide differentiation between planned and unplanned admissions. Due to the limited numbers of studies, and study heterogeneity, a more specific outcome of planned versus unplanned admissions, for example, was not possible.

Please see additional comment to clarify how hospital admission was defined, added in parentheses on Page 7, inside Table 1 “Outcome” cell:

“Prevalence or incidence studies examining emergency department use and hospitalisation (defined as an overnight stay).”

3. Results: we are not statisticians, but we would have interpreted the results of the Fisher paper differently. The results presented for Fisher et al.’s study are based on models that include interaction terms and, as such, we believe the more suitable analyses to present are found in Table 3 of the paper (which show there is a statistically significant relationship between household income and hospitalisation rates). In addition, some clarification as to why Fisher et al.’s results for two determinants - living arrangement and rurality - aren’t included in this review would be helpful.

Response 7: Thank you for this comment. Table 3 household income analysis refers to the whole population and does not stratify by the presence or absence of multimorbidity, and the same is true for living arrangement and rurality. Stratification by number of conditions appears at rows 1-3 of this table but is descriptor of the population rather than used to stratify the population into a new cohort. Only the results in Supplementary Table 6 provide results stratified by number of conditions and therefore examine interactions where hospitalisation is the outcome in groups of people with multimorbidity (where “MM group” is 2-3, and 4+) (<https://bmchealthservres.biomedcentral.com/articles/10.1186/s12913-020-06032-5#Sec22>).

4. Paper selection: could you clarify why two further papers do not meet the inclusion criteria as we feel these papers would be relevant for inclusion given this review’s inclusion criteria:

- Chung et al., (2016) The association between types of regular primary care and hospitalization among people with and without multimorbidity: A household survey on 25,780 Chinese

Response 8: Thank you for highlighting this article. It was captured with our search strategy but (in retrospect incorrectly) excluded during screening even though it does examine associations between healthcare use and marital status and household income in a population with multimorbidity. Please find additions to the paper regarding this article in Figure 1, Results, Table 2, Table 3, and Discussion sections, marked as Chung et al.

- Sum et al., (2020) Implications of multimorbidity on healthcare utilisation and work productivity by socioeconomic groups: Cross-sectional analyses of Australia and Japan

Response 9: Thank you for highlighting this article that was included in our search of titles and abstracts. This article, in Figure 5 (i)b and (ii)b

[\(https://www.ncbi.nlm.nih.gov/pmc/articles/PMC7188213/\)](https://www.ncbi.nlm.nih.gov/pmc/articles/PMC7188213/) shows parameter estimates for non-communicable disease as a predictor in a negative binomial model. The outcome is a count of the number of days spent in hospital, and the parameter is the number of extra days per extra non-communicable disease. The analysis is then stratified by household income. Because the design is a negative binomial model, there is no identification of a specific population of people with multimorbidity (because the model includes people with zero and one non-communicable disease) and therefore this study does not examine the effect of hospital use in people with multimorbidity and does not meet the inclusion criteria.

5. Clarity of findings tables.

We both struggled to understand what the main findings of the papers were in the findings tables.

Response 10: Thank you for pointing this out. We have condensed the text in Table 2, Table 3, and Table 4 to report the most salient points and improve readability.

Some suggestions are given:

- Reporting of the covariates adjusted for in each study, as these will likely influence study results.

Response 11: Please see Table 3 and Table 4 where we have included a column reporting on covariates included in each set of results.

- The results might also be more digestible if the authors included subheadings (for example study characteristics, key results for household income studies, then area deprivation studies, then rurality and so on)

Response 12: We have added additional headings as suggested in the article results section. Page 9, line 1, page 13, line 16, page 15, lines 14, 19 and 25, page 16, line 4, page 18, line 11.

- Inclusion in Table 3 of a pictorial measure so the reader can easily see the results of each study at a quick glance, rather than having to read sentences

Response 13: As per Response 10. We have explored this option at the original iteration and the pictorial presentation had the appearance of a list (as no quantitative results could be aggregated, all had different exposures, outcomes, and statistical measures). We can produce a pictorial representation if you do not think these changes are sufficient.

- Inclusion of quality ratings in the results table so reader can understand results alongside quality assessments.

Response 14: Please see updated “Methodological quality rating” column in Table 3 and Table 4.

Minor comments/questions that might require some clarification or revisions:

1. More information would be useful on the search strategy: On what dates were the database searches conducted and what did the authors do about non-English papers?

Response 15: Please see additional information to clarify these points, added to page 6, lines 9-10 and 6:

“Twelve electronic databases (MEDLINE/OVID, Embase, Global Health, PsycINFO, ASSIA, CAB Abstracts, Science Citation Index Expanded/ISI Web of Science, Scopus, Cumulative Index to Nursing and Allied Health Literature [CINAHL], Sociological Abstracts, the Cochrane Library, and OpenGrey), and reference lists, were searched for full text articles meeting eligibility criteria, published between 1st January 2000, and 21st September 2021, **which was the date the final searches took place. Searches were limited to articles written in English.**”

Please see original supplementary file Table S1 where English language limits have been applied to database searches.

2. Sometimes useful to clarify why a variable is a household level exposure: marital status for example, could be considered an individual level exposure, but possibly it’s used as a proxy for living with a partner?

Response 16: Please see additional text in discussion section, page 19, lines 27-30:

“Marital status **in this study was assumed to be a household variable. Marital status could also be considered a characteristic of the individual or of a household. It is commonly examined as a proxy for not living alone and having increased household social support in studies using routine data,^{12 39} and in this study was therefore assumed to be a household variable.**”

3. It would be useful to draw out the strengths and limitations of this review rather than the literature included in the review e.g., the fact there’s little literature and multimorbidity is differentially defined isn’t a limitation of the review itself but of the body of literature. There are a few additional strengths worth noting (e.g., considerable use of second screening, multiple people extraction data and assessments of quality using a tailored version of the NOS). There are also a few additional limitations worth noting (no meta-analysis, mix of high income and low-middle income countries in which contextual factors may hold different meanings across contexts and exclusion of non-English papers)

Response 17: Please see additional text added to the implications in the article discussion section, page 20, lines 18-21:

“No included articles examined supply factors associated with hospital use, for example area wide demand issues such as distance to services, or supply issues such as access to primary care services. These factors are likely to influence hospital use rates in people with multimorbidity and require further study.”

Please see additions to the study strengths and weaknesses sections, on page 18, line 31:

“Strengths of the study include comprehensive and systematic searching of many electronic databases, with an iteratively developed set of search terms, **extraction of data by two researchers, and** assessment of study quality with adaptation of a standard method to meet the needs of the review, and analysis using a standardised approach to narrative reporting.

Also, on page 19, lines 1-3:

“precluding meta-analysis. Studies were from both high-income and low-and-middle income countries, where contextual factors may hold different meanings, making direct comparisons across studies difficult. Finally, although multimorbidity was defined as the presence of ≥ 2 LTCs in all studies, there was a diverse range of LTCs included within counts, which will affect prevalence rates and therefore reliability of comparison between studies.¹⁹”

Reviewer: 2

Dr. Alexandru Dregan, King's College London, UK

Comments to the Author:

The authors have done an excellent work at summarising current gaps in the evidence between contextual determinants of hospitalisation with multimorbidity. Though the review does not include a meta-analysis (justified accordingly by the authors), it provides valuable information on a possible role (albeit unclear at the moment) of area-level circumstances on the growing burden of multimorbidity on the health care system. The study is methodologically appropriate and acknowledges the complexities of drawing overall conclusions from studies employing diverse measures of multimorbidity and hospitalisation. The manuscript is well-written and should be of interest to the journal audience.

My only minor comment would be to include the study country in the tables (include study exposure as a row subheading?) - might be informative with regards to some of the inconsistent evidence for countries with different distribution of wealth/income inequalities (e.g., China vs India or Canada).

Response 18. Thank you for your comments. Please see addition of study country in the “Study” columns of Table 3 and Table 4.

VERSION 2 – REVIEW

REVIEWER	Jessica Sheringham University College London
REVIEW RETURNED	16-Aug-2022

GENERAL COMMENTS	Reviewer comments on revision 1 Comments to the Author: Thank you for inviting me to review this revision. Most of our comments were addressed fully by the authors in this revision. I include here in italics only the comments where I considered more was needed. 1. There is little discussion (either in the introduction or discussion) around how demand and supply-side factors influence health and care use. For example, an individual may choose or not choose to use hospital services based on demand-side determinants like distance between their residence and a hospital, transport links, opening times of the service, and the decision of primary care clinicians or paramedics. However, decisions about emergency department use can also be affected by supply-side issues, e.g. lack of primary care access or in contrast some ED attendance/emergency hospitalisation can result from a referral to from primary care rather than being patient-initiated. Response 4: Thank you for making this useful point. We have added additional text to reference these issues in the article introduction, page 5, lines 23-26: “Additionally, factors such as geographical proximity to services are known to be associated with likelihood of ED attendance,¹⁵ and consideration of individual characteristics and variation in how services themselves operate, including supply-wide factors such as relative accessibility of primary care versus ED care are important.” Revision, reviewer comment: I lost the thread of this sentence (particularly the part “and consideration of individual characteristics and variation in how services themselves operate”) I suggest it is split into two separate ones, each dealing with one idea. There is also a typo in our comments and reproduced in the article – supply-wide should read supply-side I believe. 2. Clearer definition of the outcome: The paper’s hospitalisation outcome as we understand it is very broad. Does it include planned and unplanned hospitalisations, day visits and overnight admissions? Some clarity around the definition of the outcome of interest and breaking down as it has been used in the papers at least into elective and non-elective could have implications for the findings. Response 6: Please see Table 2, “Primary outcome(s) measures” for details of each study outcome(s). The outcome definition includes emergency department attendance and hospital admission, and most included studies did not provide differentiation between planned and unplanned admissions. Due to the limited numbers of studies, and study heterogeneity, a more specific outcome of planned versus unplanned admissions, for example, was not possible.
---

	Please see additional comment to clarify how hospital admission was defined, added in parentheses on Page 7, inside Table 1 “Outcome” cell: “Prevalence or incidence studies examining emergency department use and hospitalisation (defined as an overnight stay).” Reviewer comment: I suggest making it clear in methods/outcomes in table 1 that hospitalisations could be planned or unplanned, and noting in results/discussion that some papers did not differentiate whether hospitalisations were planned or unplanned. I think this is important, because the routes to unplanned admissions and planned admissions are/should be quite different and they have different implications for patient experience and the health care system. 5. Clarity of findings tables.  - Inclusion in Table 3 of a pictorial measure so the reader can easily see the results of each study at a quick glance, rather than having to read sentences Response 13: As per Response 10. We have explored this option at the original iteration and the pictorial presentation had the appearance of a list (as no quantitative results could be aggregated, all had different exposures, outcomes, and statistical measures). We can produce a pictorial representation if you do not think these changes are sufficient. Tables 3 and 4 are much clearer. Inclusion of quality ratings is useful. The authors could consider ordering within subcategories by quality rating, so the reader sees the highest quality papers in each category first. The reader still has to work quite hard in the table to understand which papers find an association between household/areas-level characteristics. It is not essential but I do encourage the authors to revisit the idea of a short-hand summary of the results in its own column for each paper, which might be pictorial or might be just one word or symbol with a key at the bottom e.g. “√” to represent studies where an association significant at 5% level was found, “-“ where no association was found and “√/-“ where results were mixed. Minor comments Table 4: are the result metrics prevalence or rates? I think rates. Discussion – I think there’s a typo: Principle should read Principal
--	---

VERSION 2 – AUTHOR RESPONSE

**Please update the 'Eligibility criteria' section of the abstract to mention that the literature search was limited to publications in English.*

Response 1. Please see lines 15-16 on page 2.

**Please move the PROSPERO registration number from the 'Data extraction and synthesis' section of*

the abstract to a separate section at the bottom of the abstract (ie, with 'PROSPERO registration number' as the section heading).

Response 2. Please see line 1 on page 3.

**Please change the heading 'Strengths and limitations' (after the abstract) to 'Strengths and limitations of this study'.*

Response 3. Please see line 1 on page 4.

**Please add some appropriate subheadings to the main text Methods section.*

Response 4. Please see line 5 on page 6 “**Eligibility criteria and inclusion**”, line 1 on page 8 “**Search strategy**”, line 21 on page 8 “**Quality assessment**”, and line 26 on page 8 “**Data synthesis**”.

**Please change the heading 'Data Sharing Agreement' to 'Data availability statement' and please revise the statement to indicate if any data underling your study will be made available for sharing, and via what mechanism. If not applicable, please revise the statement to read "No additional data available."*

Response 5. Please see line 12 on page 22.

**Please revise the headings 'Conflict of interests' and 'Author contributions' to 'Competing interests' and 'Contributors', respectively.*

Response 6. Please see lines 10 and 19 on page 22.

Reviewer comments.

*Revision, reviewer comment: I lost the thread of this sentence (particularly the part “and **consideration of individual characteristics and variation in how services themselves operate**”) I suggest it is split into two separate ones, each dealing with one idea. There is also a typo in our comments and reproduced in the article – supply-wide should read supply-side I believe.*

Response 7. Thank you for highlighting this. Please see changes to lines 24 and 25 of page 5 where we have split the sentence into two sections to improve readability, and the spelling mistake has been corrected.

“Additionally, factors such as geographical proximity to services are known to be associated with likelihood of ED attendance.[16] Consideration of variation in how services themselves operate, including supply-side factors such as relative accessibility of primary care versus ED care, are important.”

Reviewer comment: I suggest making it clear in methods/outcomes in table 1 that hospitalisations could be planned or unplanned, and noting in results/discussion that some papers did not differentiate whether hospitalisations were planned or unplanned. I think this is important, because the routes to unplanned admissions and planned admissions are/should be quite different and they have different implications for patient experience and the health care system.

Response 8. Thank you for making this useful point. Please see line 11 on page 6, Table 1 Inclusion Outcome cell, lines 3-4 on page 13.

Tables 3 and 4 are much clearer. Inclusion of quality ratings is useful. The authors could consider ordering within subcategories by quality rating, so the reader sees the highest quality papers in each category first.

Response 9. Thank you for making this point. We have ordered the studies within categories accordingly in Tables 3 and 4.

The reader still has to work quite hard in the table to which find an association between household/area-level characteristics. It is not essential but I do encourage the authors to revisit the idea of short-hand summary of the results in its own column for each paper, which might be pictorial or might be just one word or symbol with a key at the bottom e.g. “√” to represent studies where an association at 5% level was found, “-“ where no association was found and “√/-“ where results were mixed.

Response 10. Thank you for this useful point about clarity of results. Please see the addition of a new column to Table 3 and Table 4 “Association at 95% for ORs and IRRs”.

Minor comments

Table 4: are the result or rates? I think rates.

Response 11. Thank you for making this point. Please see addition of the term “rates” to the “Result metric” column of Table 4 where appropriate.

Discussion – I think a typo: should read Principal

Response 12. Thank you for highlighting this spelling error. Please see line 17 on page 20.